# Phosphine Oxide-Promoted Rh(I)-Catalyzed C–H Cyclization of Benzimidazoles with Alkenes

**DOI:** 10.3390/molecules28020736

**Published:** 2023-01-11

**Authors:** Mingzhen Lu, Weiwei Xu, Mengchun Ye

**Affiliations:** 1State Key Laboratory of Elemento-Organic Chemistry, College of Chemistry, Nankai University, Frontiers Science Center for New Organic Matter, Tianjin 300071, China; 2Haihe Laboratory of Sustainable Chemical Transformations, Tianjin 300192, China

**Keywords:** rhodium, C–H activation, benzimidazole, bimetal, phosphine oxide

## Abstract

Ligands play a critical role in promoting transition-metal-catalyzed C–H activation reactions. However, owing to high sensitivity of the reactivity of C–H activation to metal catalysts, the development of effective ligands has been a formidable challenge in the field. Rh(I)-catalyzed C–H cyclization of benzimidazoles with alkenes has been faced with low reactivity, often requiring very harsh conditions. To address this challenge, a phosphine oxide-enabled Rh(I)–Al bimetallic catalyst was developed for the reaction, significantly promoting the reactivity and allowing the reaction to run at 120 °C with up to 97% yield.

## 1. Introduction

Transition-metal-catalyzed C–H functionalization represents one of the most convenient methods for the construction of molecular complexity from readily available non-prefunctionalized substrates, and considerable progress has been achieved during the past several decades [1,2,3,4,5,6]. In the activation of inert C–H bonds either via electrophilic activation or via oxidative addition, transition metal catalysts often play critical roles in affecting reactivity and selectivity. In general, the catalytic ability of metal catalysts mainly relies on intrinsic properties of metals and of accompanying ligands. Therefore, the search for proper ligands to match metal catalysts is critical to most C–H functionalization reactions, and has become a long-term and challenging goal in the field [7,8,9,10]. Rh(I)-catalysis is recognized as a powerful catalytic method for C–H functionalization, and a large number of examples have been reported [11,12,13,14]. However, so far, only three types of ligand are available, namely, monophosphines [15,16,17], bisphosphines [18,19,20,21,22] and diene ligands [23,24] (Figure 1a). Owing to limited availability of ligands, some Rh(I)-catalyzed C–H functionalizations have to be run under relatively harsh conditions, resulting in difficult selectivity control and low reaction applicability. In 2001, Ellman, Bergman and co-workers reported a Rh(I)-catalyzed C–H cyclization of benzimidazoles with alkenes [25], providing an elegant route to versatile polycyclic imidazoles that widely exist in bioactive molecules (Figure 1b). However, the use of monophosphine (PCy_3_) as a ligand was not very effective, in general requiring very harsh conditions (over 160 °C and 20 h). Shortly after, to further optimize the reaction, a monophosphine (PPh_3_)-containing catalyst (RhCl(PPh_3_)) was then investigated; however, there was no obvious increase in reactivity observed, despite using a microwave-assisted technique at a higher temperature (250 °C) [26,27,28,29,30]. Switching back the monophosphine ligand with a Brønsted acid co-catalyst (PCy_3_·HCl) led to slightly improved reactivity, allowing 5 mol% Rh to be used at a slightly lower temperature (225 °C) (Figure 1b). Therefore, the search for new types of more effective ligands, instead of traditional monophosphines, is highly desirable for this reaction as well as other Rh(I)-catalyzed C–H activations. Herein, we report the use of phosphine oxide (PO) as a distinctive type of ligand to significantly promote Rh(I)-catalyzed C–H cyclization of benzimidazoles with alkenes [31,32,33], providing the desired product in up to 97% yield at 120 °C (Figure 1c). The bifunctional phosphine oxide ligand may ligate Rh(I) and Al-Lewis acid, thus activating both the Rh catalyst and benzimidazole substrates to give a big improvement in the reactivity.

## 2. Results

We recently explored a wide range of inert C–H and C–C bond activation reactions by using phosphine oxide-ligated Ni and Al bimetallic catalysis [34,35,36,37,38,39,40,41,42,43]. However, the replacement of nickel with other transition metals has been faced with big challenges, probably owing to a mismatch between the two metals, or between metal catalysts and substrates. To address this challenge, we turned our attention to the Rh(I)-catalyzed C–H cyclization of benzimidazoles with alkenes, hoping to develop a PO-ligated Rh–Al bimetallic system to improve the reactivity. As shown in Figure 2, the investigation of transition metals with Ph_2_P(O)H as a ligand showed that versatile Pd catalysts were completely ineffective (entries 1 and 2), and only Co and Rh catalysts displayed moderate to good reactivity (entries 3–9). Among various Rh catalysts, [Rh(cod)_2_]BF_4_ gave the best result, providing the cyclized product in 68% yield (entry 5). Next, a large number of phosphine oxides were then examined (entries 10–12), and only bulky Mes-DAPO afforded a better yield: 84% (entry 12). In contrast, traditional phosphines gave very low yields under the same conditions (entries 13 and 14). With Mes-DAPO as the optimal ligand, the survey of Al-Lewis acids (entries 15–19) revealed that Me_2_AlCl was the best one, providing the product in 97% yield (entry 19). Moreover, with this catalytic system, the reaction temperature can be decreased to 120 °C without observing significant loss of yield (entries 20–23). Control experiments showed that the absence of either a Rh catalyst or Al-Lewis acid led to no reaction (entries 24 and 25), and the removal of phosphine oxide gave a very low yield (12%) (entry 26), suggesting that any component of Rh, Al or PO ligand would be essential to the reactivity.

With the optimal conditions in hand, we explored the scope of benzimidazoles (Figure 3). Electron-deficient substituents such as carboester (**2b**), CF_3_ (**2c**), F (**2d**) and Cl (**2e**, **2f** and **2g**) on the phenyl ring of benzimidazoles can be highly compatible with the reaction, providing the corresponding products in 93–97% yield. In addition, electron-deficient heterocycle-bearing imidazole (**2h**) and fused aromatic ring-bearing imidazole (**2i**) were also suitable substrates, affording the corresponding products in 95% yield and 96% yield, respectively. In contrast, electron-donating substituents on the phenyl ring led to a relatively lower yield. For example, C6- and C7- monomethylated benzimidazoles gave the corresponding products in 61% yield and 52% yield, respectively (**2j** and **2k**). In addition to terminal alkenes, an internal alkene was also tolerated, providing the corresponding product in 53% yield with the increased loading of catalysts.

After the completion of the investigation of substrate scope, we turned our attention to enantioselective control with chiral phosphine oxides (Figure 4). Ellman et al. used a chiral bisphosphine-Rh(I) system to obtain a β-chiral stereocenter, while the construction of a more sterically hindered α-chiral stereocenter via Rh(I) catalysis still remains an elusive challenge. We examined a wide range of chiral phosphine oxides and found that only bulky phosphine oxide (**L_1_**) bearing a flexible chain gave 27% ee, suggesting that chiral phosphine oxide–Rh–Al would be a feasible enantioselective catalytic system. Surprisingly, a cyclohexane-1,2-diamine-derived phosphine oxide–Co–Al system provided better yield and ee (65% yield and 45% ee) in the same reaction, suggesting that a phosphine oxide ligand could also be a potential chiral ligand for Co(I) catalyzed C–H activation reactions.

## 3. Discussion

To gain more insights into the reaction, some mechanistic experiments were conducted (Figure 5a). A deuterium-labeling experiment showed that C2–D was distributed in several positions in the product, suggesting that the migratory insertion of an alkene into the Rh–H species could be a reversible step.

In addition, parallel experiments revealed a low kinetic isotope effect (*k*_H_/*k*_D_ = 1.44), implying that the C–H activation step may not be involved in the rate-determining step. On the basis of these results and previous studies, a plausible mechanism is proposed in Figure 5b: the combination of the PO ligand, Rh and Al-Lewis acid in situ formed a bimetallic catalyst, which then coordinated to form benzimidazole at the Al terminus. Then, the Rh was directed to activate C2–H via oxidative addition, followed by migratory insertion of an alkene and reductive elimination to generate the desired product. Although the current result is not significantly exciting, it demonstrates for the first time that phosphine oxides can be used as new and effective ligands for Rh(I)- or Co(I)-catalyzed C–H activation reactions, in which proper ligands are still quite scarce. Moreover, phosphine oxides allow co-catalysis of Lewis acid metals to assist Rh(I) or Co(I) catalysis, providing more options for catalyst design. 

## 4. Materials and Methods

### 4.1. Typical Procedure for Rh-Catalyzed C–H Cyclization

To a 15 mL oven-dried tube, we added Mes-DAPO (6.8 mg, 10 mol%), [Rh(cod)_2_]BF_4_ (8.1 mg, 10 mol%), benzimidazole derivative (0.2 mmol) and dry degassed toluene (2.0 mL) under N_2_ atmosphere. Then, AlMe_2_Cl (1.0 M/hexane, 40 μL, 20 mol%) was added, and the tube was sealed. The reaction mixture was heated at 120 °C for 3 h and then cooled to room temperature. The resulting solution was quenched with 5% EDTA disodium salt solution (2 mL), filtered through silica gel (EtOAc as the eluent) and concentrated in vacuo. The residue was further purified with flash column chromatography on silica gel (eluting with EtOAc/hexanes) to give the pure product (see Appendix A).

#### 4.1.1. 3-Methyl-2,3-dihydro-1H-benzo[d]pyrrolo[1,2-a]imidazole (**2a**) [39]

**^1^H NMR** (400 MHz, CDCl_3_) *δ* 7.80–7.64 (m, 1H), 7.31–7.16 (m, 3H), 4.18–4.07 (m, 1H), 4.06–3.91 (m, 1H), 3.46–3.27 (m, 1H), 2.94–2.81 (m, 1H), 2.32–2.22 (m, 1H), 1.48 (d, *J* = 7.0 Hz, 3H). **^13^C NMR** (100 MHz, CDCl_3_) *δ* 164.5, 148.7, 132.4, 121.9, 121.7, 119.7, 109.6, 42.0, 35.4, 31.0, 18.1.

#### 4.1.2. Methyl-3-methyl-2,3-dihydro-1H-benzo[d]pyrrolo[1,2-a]imidazole-6-carboxylate (**2b**) [39]

**^1^H NMR** (400 MHz, CDCl_3_) *δ* 8.39 (s, 1H), 7.93 (d, *J* = 8.4 Hz, 1H), 7.27 (d, *J* = 8.3 Hz, 1H), 4.18–4.11 (m, 1H), 4.05–3.97 (m, 1H), 3.91 (s, 3H), 3.43–3.31 (m, 1H), 2.96–2.85 (m, 1H), 2.35–2.24 (m, 1H), 1.48 (d, *J* = 7.0 Hz, 3H). **^13^C NMR** (100 MHz, CDCl_3_) *δ* 167.9, 166.3, 148.4, 135.7, 123.8, 122.0, 109.2, 77.5, 52.1, 42.1, 35.4, 31.1, 18.0.

#### 4.1.3. 3-Methyl-6-(trifluoromethyl)-2,3-dihydro-1H-benzo[d]pyrrolo[1,2-a]imidazole (**2c**) [39]

**^1^H NMR** (400 MHz, CDCl_3_) *δ* 7.96 (s, 1H), 7.45 (d, *J* = 8.4 Hz, 1H), 7.35 (d, *J* = 8.4 Hz, 1H), 4.24–4.12 (m, 1H), 4.12–3.99 (m, 1H), 3.49–3.34 (m, 1H), 2.99–2.87 (m, 1H), 2.38–2.27 (m, 1H), 1.50 (d, *J* = 7.1 Hz, 3H). **^13^C NMR** (100 MHz, CDCl_3_) *δ* 166.6, 148.1, 134.4, 126.4, 124.4, 124.1, 123.7, 119.1, 119.0, 117.3, 117. 3, 109.9, 42.2, 35.4, 31.2, 18.0. **^19^F NMR** (376 MHz, CDCl_3_) *δ* −60.5.

#### 4.1.4. 6-Fluoro-3-methyl-2,3-dihydro-1H-benzo[d]pyrrolo[1,2-a]imidazole (**2d**) [39]

**^1^H NMR** (400 MHz, CDCl_3_) *δ* 7.36 (d, *J* = 9.7 Hz, 1H), 7.16 (dd, *J* = 8.7, 4.6 Hz, 1H), 6.99–6.85 (m, 1H), 4.15–4.05 (m, 1H), 4.02–3.92 (m, 1H), 3.41–3.25 (m, 1H), 2.95–2.80 (m, 1H), 2.32–2.21 (m, 1H), 1.46 (d, *J* = 7.0 Hz, 3H). **^13^C NMR** (100 MHz, CDCl_3_) *δ* 166.0, 160.3, 158.0, 158.0, 149.1, 149.0, 129.0, 110.1, 109.8, 109.7, 109.6, 105.8, 105.5, 42.2, 35.4, 31.3, 18.0. **^19^F NMR** (376 MHz, CDCl_3_) *δ* −121.6.

#### 4.1.5. 6,7-Dichloro-3-methyl-2,3-dihydro-1H-benzo[d]pyrrolo[1,2-a]imidazole (**2e**) [39]

**^1^H NMR** (400 MHz, CDCl_3_) *δ* 7.72 (s, 1H), 7.32 (s, 1H), 4.12–4.01 (m, 1H), 4.00–3.88 (m, 1H), 3.40–3.26 (m, 1H), 2.95–2.82 (m, 1H), 2.34–2.22 (m, 1H), 1.45 (d, *J* = 7.2 Hz, 3H). **^13^C NMR** (100 MHz, CDCl_3_) *δ* 166.7, 148.0, 131.6, 125.7, 125.6, 120.7, 110.9, 42.2, 35.3, 31.1, 17.9.

#### 4.1.6. 7-Chloro-3-methyl-2,3-dihydro-1H-benzo[d]pyrrolo[1,2-a]imidazole (**2f**) [39]

**^1^H NMR** (400 MHz, CDCl_3_) *δ* 7.58 (d, *J* = 8.3 Hz, 1H), 7.27 (s, 1H), 7.15 (d, *J* = 8.1 Hz, 1H), 4.28–4.07 (m, 1H), 4.07–3.90 (m, 1H), 3.58–3.24 (m, 1H), 3.09–2.78 (m, 1H), 2.44–2.19 (m, 1H), 1.47 (d, *J* = 6.1 Hz, 3H). **^13^C NMR** (100 MHz, CDCl_3_) *δ* 165.4, 147.2, 132.9, 127.6, 122.3, 120.4, 109.8, 42.1, 35.3, 31.0, 18.0.

#### 4.1.7. 6-Chloro-3-methyl-2,3-dihydro-1H-benzo[d]pyrrolo[1,2-a]imidazole (**2g**) [39]

**^1^H NMR** (400 MHz, CDCl_3_) *δ* 7.76 (s, 1H), 7.35–7.22 (m, 2H), 4.26–4.17 (m, 1H), 4.14–4.02 (m, 1H), 3.56–3.36 (m, 1H), 3.04–2.94 (m, 1H), 2.47–2.34 (m, 1H), 1.57 (d, *J* = 7.0 Hz, 3H). **^13^C NMR** (100 MHz, CDCl_3_) *δ* 165.9, 149.3, 131.0, 127.4, 122.4, 119.5, 110.3, 42.2, 35.3, 31.2, 18.0.

#### 4.1.8. 6-Methyl-7,8-dihydro-6H-pyrrolo[2′,1′:2,3]imidazo[4,5-b]pyridine (**2h**) [39]

**^1^H NMR** (400 MHz, CDCl_3_) *δ* 8.25 (d, *J* = 4.8 Hz, 1H), 7.93 (d, *J* = 8.0 Hz, 1H), 7.15 (dd, *J* = 8.0, 4.9 Hz, 1H), 4.34–4.24 (m, 1H), 4.16–4.06 (m, 1H), 3.50–3.28 (m, 1H), 2.99–2.81 (m, 1H), 2.35–2.25 (m, 1H), 1.49 (d, *J* = 7.0 Hz, 3H). ^1**3**^**C NMR** (100 MHz, CDCl_3_) *δ* 165.8, 146.1, 143.0, 140.8, 127.1, 117.9, 41.5, 35.3, 31.6, 17.8.

#### 4.1.9. 3-Methyl-2,3-dihydro-1H-naphtho[2,3-d]pyrrolo[1,2-a]imidazole (**2i**) [39]

**^1^H NMR** (400 MHz, CDCl_3_) *δ* 8.15 (s, 1H), 7.96 (s, 1H), 7.88 (s, 1H), 7.62 (s, 1H), 7.43–7.31 (m, 2H), 4.18–4.06 (m, 1H), 4.05–3.93 (m, 1H), 3.46–3.30 (m, 1H), 2.94–2.81 (m, 1H), 2.36–2.23 (m, 1H), 1.56–1.44 (m, 3H). **^13^C NMR** (100 MHz, CDCl_3_) *δ* 168.8, 149.0, 133.3, 130.2, 128.6, 127.4, 124.2, 123.4, 116.3, 105.2, 42.0, 35.3, 31.3, 17.9.

#### 4.1.10. 3,7-Dimethyl-2,3-dihydro-1H-benzo[d]pyrrolo[1,2-a]imidazole (**2j**) [39]

**^1^H NMR** (400 MHz, CDCl_3_) *δ* 7.57 (d, *J* = 8.2 Hz, 1H), 7.08 (s, 1H), 7.02 (d, *J* = 8.3 Hz, 1H), 4.14–4.02 (m, 1H), 4.01–3.88 (m, 1H), 3.43–3.24 (m, 1H), 2.91–2.81 (m, 1H), 2.46 (s, 3H), 2.31–2.19 (m, 1H), 1.46 (d, *J* = 7.0 Hz, 3H). **^13^C NMR** (100 MHz, CDCl_3_) *δ* 164.0, 146.5, 132.5, 131.8, 123.2, 119.1, 109.6, 41.8, 35.4, 31.0, 21.8, 18.1. 

#### 4.1.11. 3,8-Dimethyl-2,3-dihydro-1H-benzo[d]pyrrolo[1,2-a]imidazole (**2k**) [39]

**^1^H NMR** (400 MHz, CDCl_3_) *δ* 7.52 (d, *J* = 8.0 Hz, 1H), 7.07 (t, *J* = 7.6 Hz, 1H), 6.92 (d, *J* = 7.1 Hz, 1H), 4.47–4.29 (m, 1H), 4.27–4.10 (m, 1H), 3.39–3.18 (m, 1H), 2.91–2.80 (m, 1H), 2.57 (s, 3H), 2.33–2.16 (m, 1H), 1.46 (d, *J* = 7.0 Hz, 3H). **^13^C NMR** (100 MHz, CDCl_3_) *δ* 164.3, 148.3, 131.8, 123.2, 121.9, 120.7, 117.3, 44.2, 35.5, 30.6, 18.1, 17.1.

#### 4.1.12. 3-Propyl-2,3-dihydro-1H-benzo[d]pyrrolo[1,2-a]imidazole (**2l**) [39]

**^1^H NMR** (400 MHz, CDCl_3_) *δ* 7.80–7.66 (m, 1H), 7.32–7.27 (m, 1H), 7.24–7.16 (m, 2H), 4.17–4.08 (m, 1H), 4.06–3.93 (m, 1H), 3.39–3.17 (m, 1H), 2.91–2.76 (m, 1H), 2.38–2.29 (m, 1H), 2.05–1.95 (m, 1H), 1.66–1.49 (m, 3H), 0.99 (t, *J* = 7.0 Hz, 3H). **^13^C NMR** (100 MHz, CDCl_3_) *δ* 164.0, 148.7, 132.3, 121.9, 121.7, 119.8, 109.6, 42.1, 36.1, 35.5, 33.3, 20.7, 14.3.

### 4.2. Procedure for Enantioselective C–H Cyclization

#### 4.2.1. Procedure for Enantioselective Rh-Catalyzed C–H Cyclization

To an oven-dried tube (15 mL), we added ligand **L1** (11.1 mg, 10 mol%), [Rh(cod)_2_]BF_4_ (8.1 mg, 10 mol%), benzimidazole derivative (0.2 mmol) and dry degassed toluene (2.0 mL) under N_2_ atmosphere. Then AlMe_2_Cl (1.0 M/hexane, 40 μL, 20 mol%) was added, and the tube was sealed. The reaction mixture was heated at 120 °C for 3 h and then cooled to room temperature. The resulting solution was quenched with 5% EDTA disodium salt solution (2 mL), filtered through silica gel (EtOAc as the eluent) and concentrated in vacuo. The residue was further purified with flash column chromatography on silica gel (eluting with EtOAc/hexanes) to give the pure product. (*R*)-3-Methyl-2,3-dihydro-1*H*-benzo[*d*]pyrrolo[1,2-a]imidazole. HPLC condition: Chiralpak IC column, *n*-hexane/*i*-PrOH = 85:15, 1.0 mL/min, 254 nm, *t_r_*_-major_ = 15.6 min, 27% ee. ${\left[\alpha \right]}_{D}^{28}$ + 2.64 (c 0.5, CHCl_3_).

#### 4.2.2. Procedure for Enantioselective Co-Catalyzed C–H Cyclization

To an oven-dried tube (15 mL), we added ligand **L2** (10.8 mg, 10 mol%), PCy_3_ (5.6 mg, 10 mol%), CoCl_2_ (2.6 mg, 10 mol%), Zn (6.5 mg, 50 mol%), benzimidazole derivative (0.2 mmol, 1 equiv.) and dry degassed toluene (2.0 mL) under N_2_ atmosphere. Then, AlMe_3_ (1.0 M/hexane, 80 μL, 40 mol%) was added, and the tube was sealed. The reaction mixture was heated at 120 °C for 3 h and then cooled to room temperature. The resulting solution was quenched with 5% EDTA disodium salt solution (2 mL), filtered through silica gel (EtOAc as the eluent) and concentrated in vacuo. The residue was further purified with flash column chromatography on silica gel (eluting with EtOAc/hexanes) to give the pure product. (*S*)-3-Methyl-2,3-dihydro-1*H*-benzo[*d*]pyrrolo[1,2-a]imidazole. HPLC condition: Chiralpak IC column, *n*-hexane/*i*-PrOH = 85:15, 1.0 mL/min, 254 nm, *t_r_*_-major_ = 22.1 min, 45% ee. ${\left[\alpha \right]}_{D}^{28}$ −6.80 (c 0.5, CHCl_3_).

### 4.3. Procedure for Mechanistic Experiments

The deuterium-labeling experiment and parallel experiments were set up following the general procedure by using **1a** or *d*-**1a** as substrates, respectively. Aliquots were taken at proper intervals. The yield was determined using ^1^H NMR with CH_2_Br_2_ as an internal standard. Data points represent the average of two runs.

## 5. Conclusions

A phosphine oxide-ligated Rh–Al bimetal-catalyzed selective C2–H cyclization of benzimidazoles with alkenes was developed, providing a series of polycyclic imidazoles in up to 97% yield under relatively mild conditions (120 °C and 3 h). This work demonstrated that the phosphine oxide ligand is a distinctive type of ligand for Rh(I) catalysis compared with traditional monophosphines, bisphosphines and diene ligands, and a wide range of applications may be expected in the future.

## Data Availability

Not applicable.

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
