# Peer review of "Phosphine Oxide-Promoted Rh(I)-Catalyzed C–H Cyclization of Benzimidazoles with Alkenes"

_molecules, 2023, doi:10.3390/molecules28020736_

Round 1

Reviewer 1 Report

               The topic of the Manuscript „Phosphine Oxide-Promoted Rh(I)-Catalyzed C–H Cyclization of Benzimidazoles with Alkenes“ is relevant and of interest to the audience of this journal.

               The content of this paper is technically accurate, however the abstract is very short and insufficiently informative. The introduction provides the necessary background information and the references are appropriate. The results of the analysis are correctly interpreted, however the dicsussion is very short. Tables and figures are appropriate. Supplementary material was not available.

               In order to improve this manuscript, the discussion needs to be more detailed and the abstract more informative.

Author Response

Response to Reviewer #1: Great thanks for the helpful comments

  1. “In order to improve this manuscript, the discussion needs to be more detailed and the abstract more informative.”

Response: More detailed information has been added to the abstract and the discussion of the manuscript. 

Reviewer 2 Report

The manuscript describe a Rh-catalyzed cyclization using Mes-DAPO as ligand, and AlMe2Cl as the necessary Lewis acid, which is a complementary research of the author’s previous report (J. Am. Chem. Soc. 2018, 140, 5360−5364). The scientific novelty as well as the overall quality of the manuscript is suitable to be publish in Molecules, but there is no supporting information for all needed spectra provided. The author therefore should submit the corresponding SI for further review before the final acceptance.

Other suggestions:

For a better understanding, each intermediate for the catalytic cycle in Scheme 5b should be labelled as “A, B, C…”.

Author Response

Great thanks for the helpful comments

  1. “The scientific novelty as well as the overall quality of the manuscript is suitable to be publish in Molecules, but there is no supporting information for all needed spectra provided. The author therefore should submit the corresponding SI for further review before the final acceptance.”

Response: The supporting information has been attached.

  1. “For a better understanding, each intermediate for the catalytic cycle in Scheme 5b should be labelled as “A, B, C…”.”

Response: Intermediates in Scheme 5b have been labelled as A, B and C.

Reviewer 3 Report

The authors have reported phosphine oxide-promoted Rh(I)-catalyzed C–H cyclization of benzimidazoles with alkenes.  The author has mentioned that the reaction generally suffers with low reactivity and often requires harsh conditions citing earlier methods. In the present work, they have accomplished the C-H cyclization of benzimidazoles with alkenes promoted by phosphine oxide (Mes-DAPO) to yield in 52-97% yields as demonstrated for over 12 examples.  The reaction was well demonstrated with terminal alkynes and also with internal alkyne (albeit low yield). The work looks reasonable to publish but the author needs to further rewrite to justify the novelty of their present work.  As the similar transformation is well known in J. Am. Chem. Soc. 2020, 142, 3, 1200–1205 even chiral version is known with internal alkynes see. J. Am. Chem. Soc.2018, 140, 16, 5360–5364 wherein secondary phosphine oxide ligand was used. In many cases the temperatures are beyond 100 oC and hence matching temperature may not be right criteria (as the authors do at 120 and earlier 160 is reported).

If the authors can add few more examples with intermolecular cyclization reactions it may increase the viability for a wide readership.  Similarly they may have to stress the novelty of the present protocol.  If these are addressed, then the manuscript may be considered for publication.

Author Response

Great thanks for the helpful comments

  1. “The work looks reasonable to publish but the author needs to further rewrite to justify the novelty of their present work. As the similar transformation is well known in J. Am. Chem. Soc.2020, 142, 3, 1200–1205 even chiral version is known with internal alkynes see. J. Am. Chem. Soc.2018, 140, 16, 5360–5364 wherein secondary phosphine oxide ligand was used. In many cases the temperatures are beyond 100 oC and hence matching temperature may not be right criteria (as the authors do at 120 and earlier 160 is reported). If the authors can add few more examples with intermolecular cyclization reactions it may increase the viability for a wide readership. Similarly, they may have to stress the novelty of the present protocol. If these are addressed, then the manuscript may be considered for publication.”

Response: Either intramolecular or intermolecular CH alkylation of benzimidazoles with alkenes has been investigated by using many other metals such as Pd, Ni, Sc and Cu with proper ligands, and excellent yield and even high enantioselectivity have been achieved. Compared with these progress, Rh(I)-catalyzed version has not been developed very well, mainly owing to the lack of effective ligands. Therefore, we hoped to use the reaction as a model reaction to search effective ligands for Rh(I)-catalyzed CH activation. Finally, phosphine oxides were identified as new and effective ligands for Rh(I) or Co(I) catalyzed C–H activation reactions, providing high reactivity albeit with moderate enantioselectivity. Although the result is not exciting enough so far, it would open an avenue to the future development of Rh(I) or Co(I) catalyzed C–H activation reactions. To more clearly describe the novelty of the current work, the abstract, introduction and discussion has been rewritten and some detailed information has been added.

Round 2

Reviewer 1 Report

The manuscript has been improved according to suggestions.

Author Response

no requests for new revisions

Reviewer 2 Report

The authors have submitted the SI and well addressed the requests from the referees. The manuscript has been substantially improved and the current version is suitable for publication.

Author Response

no requests for new revisions